# Micronekton indicators evolution based on biophysically defined provinces

Authors: Sarah Albernhe[a, b], Thomas Gorgues[b], Olivier Titaud[a], Patrick Lehodey[c, d], Christophe Menkes[e], Anna Conchon[a]

[a] Collecte Localisation Satellites, 8-10 rue Hermès, Ramonville Sant Agne 31520, France

[b] Univ Brest, CNRS, Ifremer, IRD, Laboratoire d'Océanographie Physique et Spatiale (LOPS), IUEM, F29280, Plouzané, France.

[c] Mercator Ocean International, 2 Av. de l'Aérodrome de Montaudran, 31400, Toulouse, France

[d] Pacific Community, Oceanic Fisheries Programme, Noumea, New Caledonia

[e] ENTROPIE, IRD, Univ. de La Réunion, CNRS, Ifremer, Univ. de la Nouvelle-Calédonie, BP A5, 98848 Nouméa, New Caledonia

*Correspondence to:* Sarah Albernhe (salbernhe@groupcls.com)

**Abstract.** Micronekton are mid-trophic marine organisms characterized by a size range of 2 to 20 cm, gathering a wide diversity of taxa (crustaceans, fish, molluscs). They are responsible for an important active carbon export to the deep ocean because of their diel vertical migrations and constitute the main prey for pelagic predators. A new method has been proposed in the literature to define provinces that identify micronekton functioning patterns based on environmental variable. Following this methodology, we define homogeneous provinces using environmental variables computed from Copernicus Marine Service products. These provinces represent a relevant way to define regions of interest, offering a regional scope of study for micronekton indicators and their evolution in time. In this study, we observe the evolution of the provinces in time from 1998 to 2023, to account for the decadal to climatic variability. We focus on the variations in surface area and average latitude of each province. We observe a global shrinking of productive provinces and polar provinces, in favor of equatorial and tropical provinces expansion. Additionally, tracking the geographical changes of the provinces over time shows that most are shifting toward the poles.

40

**1 Introduction**

The intermediate level of the oceanic food web is constituted by a group of marine organisms called micronekton, understudied yet, but garnering increasing attention. This key component of marine ecosystems characterized by organisms in a size range from 2 to 20 cm contains a wide diversity of taxa such as crustaceans, fish, molluscs and gelatinous (Brodeur et al., 2004; Escobar-Flores et al., 2019). Micronekton mostly feed on zooplankton and are the main prey of marine large predators, some of which are of crucial economic importance (e.g. tunas, Bell et al., 2015; Terawasi et al., 2017; McCluney et al., 2019). In addition to their role as prey for commercially exploited top predators (Young et al., 2015), mesopelagic micronekton itself could become a valuable resource for fisheries, due to the increasing demand for fishmeal for aquaculture (St. John et al., 2016; Gatto et al., 2023). Another aspect of micronekton worthy of interest lies in its migratory behaviour, which impact global carbon export (Pinti et al. 2021; Buesseler et al., 2022) by actively transporting and sequestrating carbon beneath the mixed layer (Bianchi et al., 2013; Boyd et al., 2019; Gorgues et al., 2019).

Therefore, estimating micronekton biomass is a major concern for fisheries management and climate regulation. Direct observations of micronekton primarily rely (i) on ship-borne acoustic measurements, which does not provide yet a reliable representation of the micronekton biomass (McGehee et al., 1998; Kloser et al., 2002), and (ii) on trawl sampling, which is susceptible to biases due for example to species avoidance (Kaartvedt et al., 2012) and has a coarse sampling. Numerical models, such as the Spatial Ecosystem and Population Dynamics Model – Low and Mid Trophic Levels (SEAPODYM-LMTL: Lehodey et al., 2010; 2015; Conchon, 2016) are complementary tools for studying micronekton biomass. Indeed, by simulating micronekton dynamics based on key biological and physical processes (such as growth, recruitment, mortality and environmental influences), these models provide a continuous representation of micronekton biomass across space and time. This helps fill observational gaps, enabling the analysis of large-scale patterns, the simulation of future scenarios, and ultimately a better understanding of the mesopelagic ecosystem.

One approach to quantify and characterize the mid-trophic level populations is the definition of homogeneous provinces. Longhurst was the pioneer and defined a static vision of biogeographical provinces based on chlorophyll fields (Longhurst 1995; 2007). Various combinations of features have been used to create accurate definitions of provinces for each field: environmental features such as the distribution of species (Costello et al., 2017) and phytoplankton species assemblages (Elizondo et al., 2021), biogeographic insights from multi-expertise discussions (Sutton et al., 2017), and fisheries-related data, such as catch per unit of effort of commercial fisheries (Reygondeau et al., 2012). Acoustic-based regionalization is also explored, using environmental drivers' classification to model backscattering characteristics (Proud et al., 2017), or recently partitioning acoustic data according to the vertical structure of sound-scattering mid-trophic biomass (Ariza et al., 2022).

Complementing these approaches, Albernhe et al. (2024) proposed a new methodology for regionalizing the global ocean into biophysical provinces based on environmental variables. Since the present study builds upon Albernhe et al. (2024), we detail the main and key findings of the prior study in the following sentences. The ambition of the prior study Albernhe et al. (2024) was to identify micronekton homogeneous functioning patterns using a parsimonious set of biophysical variables that are known to have an impact on micronekton biomass (epipelagic layer temperature, stratification of the mesopelagic ocean temperature, and net primary production (NPP)). Clustering these variables results in a global classification of six distinct biomes (tropical, subtropical, eastern boundary coastal upwelling systems, oceanic mesotrophic systems, sub-polar and polar biomes). The authors also defined a monthly time series of biomes for the 1998-2019 time period. From these large biomes, provinces are derived as biomes' sub-divisions at the scale of ocean basin and hemisphere. A characterization of these provinces with simulated micronekton from SEAPODYM-LMTL model outputs identifies biomes-specific relations between micronekton biomasses and the environmental variables used in the clustering. Additionally, biomes-specific vertical structures are indicated by ratios of modelled micronekton functional groups (i.e., groups of micronekton with specific migratory behaviour, and specific depth habitat). Boundaries between provinces have also been validated using acoustic data. With demonstrated accuracy in homogeneous micronekton characteristics, these provinces enable the gathering and extrapolation of the few available observation data of micronekton over large homogeneous areas. This could benefit the exploration of the micronekton spatio-temporal variability within global or regional datasets.

In the present study, we focus on provinces' features, such as surface area and positional changes, which serve as valuable indicators providing insights into the evolution of ecosystem structure over time, both globally and regionally. Following Albernhe et al. (2024)'s methodology, we define in the present study an annual time series of biophysical provinces from 1998 to 2023. We observe the evolution of two geographical indicators: the surface area and the average latitude of each province.

## 2 Material and methods

### 2.1 Environmental variables and biophysical clustering

We define a time series of biophysical provinces from 1998 to 2023 following Albernhe et al. (2024)'s approach. The latter publication offers a methodology for global ocean regionalization based on environmental variables, with no gaps and no overlaps, displaying homogeneous biophysical characteristics. While the overall methodology is detailed in Albernhe et al. (2024), we outline the different steps of the method below to ensure this study is comprehensive and self-contained.

We consider three environmental variables, that are known to have an impact on micronekton: the mean temperature in the epipelagic layer, the temperature gradient between the epi and the meso-pelagic layers, as an index of the stratification (hereafter referred to as 'stratification'), and the integrated NPP. The pelagic layers mentioned are defined as in SEAPODYM-LMTL. These variables are computed from the biological and physical Copernicus Marine Service datasets of the product *Global Ocean low and mid trophic levels biomass content hindcast*, GLOBAL_MULTIYEAR_BGC_001_033 (1/12° horizontal resolution, product ref01, table 1). In the product, the weekly 3D temperature fields come from the GLORYS12V1 simulation. NPP and the associated euphotic depth are computed using the Vertically Generalized Production Model (VGPM) of Behrenfeld and Falkowski (1997) which is based on the Satellite Observations reprocessed Global Ocean Chlorophyll product. The spatial domain of our study is restricted to area where the depth of the water column supports the existence of all three pelagic layers as defined in SEAPODYM-LMTL (i.e. roughly 1000m deep, See Material and Method section of Albernhe et al. (2024)). Consequently, shallow coastal areas are excluded from this analysis. Annual time series of these three variables (i.e., epipelagic layer temperature, stratification and NPP), spatially averaged on a global scale, are provided in the Supplementary Material (Figure S1). This illustrates how the global mean values of temperature, stratification, and NPP fluctuate over time, reflecting interannual variability and decadal trends at the global scale.

As described in Albernhe et al. (2024), a Principal Component Analysis (PCA) (Hotelling, 1933) is performed on the three environmental variables mentioned above (i.e., epipelagic layer temperature, stratification and NPP), producing empirical orthogonal functions that strongly mirror those identified in Albernhe et al. (2024). We selected the two principal components that explain the most variance, accounting for 98,1% of the variance (68,2% and 29,9% for the first and second PCA respectively).

Then, a clustering is performed on the PCA two principal components, hereafter referred to as "biophysical clustering". Our goal is to define homogeneous biophysical biomes by detecting intrinsic patterns or structures within the data, without relying on any predefined clustering assumptions. Thus, the biophysical clustering is performed using the unsupervised k-means machine learning algorithm (Lloyd, 1957; Pedregosa et al., 2011), which partitions the observations into k homogeneous clusters (See Material and Method section of Albernhe et al. (2024)). In Albernhe et al. (2024), we identified six clusters (k = 6) to classify global-scale environmental data, effectively distinguishing biophysical biomes. In this study, the different metrics used to determine the optimal number of clusters do not exhibit a strongly pronounced pattern. One suggests that k = 5 could be a suitable choice, albeit not with strong certainty. To ensure consistency with Albernhe et al. (2024), we maintain k = 6, allowing us to build upon our previous findings on micronekton biomass and vertical structure. The clusters derived from the clustering define six homogeneous biomes on a global scale, hereafter referred to as "biophysical biomes".

First, the training phase of k-means algorithm is applied to time-averaged 1/12-degree datasets from 1998 to 2023. This process defines static reference biophysical biomes, representing the average state of the ocean over the entire period. After the training phase, the clustering model parameters are estimated, and we can use this model to make predictions on other data. Then, the prediction phase of k-means algorithm is applied on monthly data over the

same time period (1998-2023) (See Material and Method section of Albernhe et al. (2024)). This produces a monthly time series of biophysical biomes.

The six biophysical biomes obtained from the clustering of environmental data characterize homogeneous environmental regimes on a global scale. Since similar oceanographic regimes occur in multiple locations, biophysical biomes extend across various ocean basins. In this study, we also delineate "provinces" as subdivisions of biomes at the scale of ocean basins and hemispheres that have been shown to be characterized by stable biophysical drivers and potential taxonomic identity (Spalding et al., 2012; Sutton et al., 2017; Albernhe et al., 2024). This subdivision of each of the six biophysical biomes results in the definition of 27 provinces, establishing regional frameworks for studying micronekton.

## 2.2 Trends identification

The aim of this study is to analyze the evolution of the provinces in time from 1998 to 2023. The biophysical data described in the previous section are available at a monthly resolution and monthly provinces are derived through clustering, in order to follow Albernhe et al. (2024)'s methodology. While provinces are resolved monthly, our analysis focuses on decadal to climatic trends aiming to identify long-term patterns in their evolution. To study the temporal variability and identify potential trends over the 26 years, we consider the annual time series. We calculate indicators based on the monthly definition of provinces and then compute the annual averages of these indicators. We document the evolution of two geographical indicators: the surface area and the average latitude, for each province. The average latitude diagnostic has been designed to assess a potential poleward displacement of certain provinces (see Hastings et al., 2020; Pinsky et al. 2020 and references therein).

To evaluate the evolution of surface area over time, our approach is based on a simple linear regression model applied to the annual surface area (in km²) of each province from 1998 to 2023. We analyze the slope of the regression (in km²/year), to account for the direction and first-order magnitude of variability. Rather than directly comparing the years 1998 and 2023 to quantify the variation between these dates (which would assume that surface areas for these two years perfectly align with a statistically significant linear trend), we project the equivalent evolution over 26 years based on the slope of the regression (i.e., 26 × slope). From this projected variation, we compute the percentage change in surface area over 26 years (in %) relative to the surface area at the start of the time series (year 1998).

Similarly, to track the poleward drift of provinces over time, our approach is based on a simple linear regression model applied to the average latitude of each province of each province from 1998 to 2023. We analyze the slope of the regression (in degrees poleward/year), to account for the direction and first-order magnitude of variability. The 'degree poleward' unit that we use for this diagnostic is associated with degree N for provinces in the northern hemisphere, and degree S for provinces in the southern hemisphere. Thus, provinces belonging to the equatorial Biome 1 (provinces 101, 102 and 103) are not considered in this diagnostic because of their equatorial position. Rather than directly comparing the years 1998 and 2023 to quantify the variation between these dates, we project the equivalent evolution over 26 years based on the slope of the regression (in degree poleward), following the same approach as described for the surface area metric.

To track the poleward drift of provinces over time, we analyze the slope (in degrees poleward/year) of a linear regression model based on the average latitude of each province throughout the annual time series from 1998 to 2023. The 'degree poleward' unit that we use for this diagnostic is associated with degree N for provinces in the northern hemisphere, and degree S for provinces in the southern hemisphere. Thus, provinces belonging to the equatorial Biome 1 (provinces 101, 102 and 103) are not considered in this diagnostic because of their equatorial position. Derived from the linear regressions, we estimate the poleward variation trend over the 26 years for each province (in degree poleward), based on the difference between the first and last point of the regression (respectively matching 1998 and 2023).

Recapitulative tables for each of these two metrics are provided in the supplementary material (Table S3 for surface area and Table S5 for mean latitude). These tables present, for each province, the trend from the linear regression model, the total variation over the 26 years, and the coefficient of determination ($R^2$) for each regression. $R^2$ is a statistical measure that evaluates the degree of fit between the observed values and the linear regression model, allowing the statement of statistically significant linear trends. A 26-year period is too short to detect statistically significant trends in such biophysical features. Due to interannual variability, $R^2$ values are not expected to be

close to 1, which would indicate statistical significance of the linear trends. The purpose of the linear regressions
is to identify the direction and relative magnitude of the trends, rather than to confirm their statistical significance.
Caution must be taken while considering such trends. Thus, scatter plots of the annual time series for surface area
(Figure S2) and mean latitude (Figure S4), with the corresponding linear regression, are provided in the
supplementary material for each province. These plots allow for direct observation of the time series.

## 3 Results

### 3.1 Biophysical provinces definition

To define the homogeneous biophysical biomes, we perform a clustering on the two principal components
generated by the PCA performed on the three environmental variables (i.e. epipelagic layer temperature,
stratification and NPP). From the learning phase of the clustering algorithm, six static reference biophysical biomes
(Figure 1) are defined on a global scale, representing the average state of the ocean over the entire period.

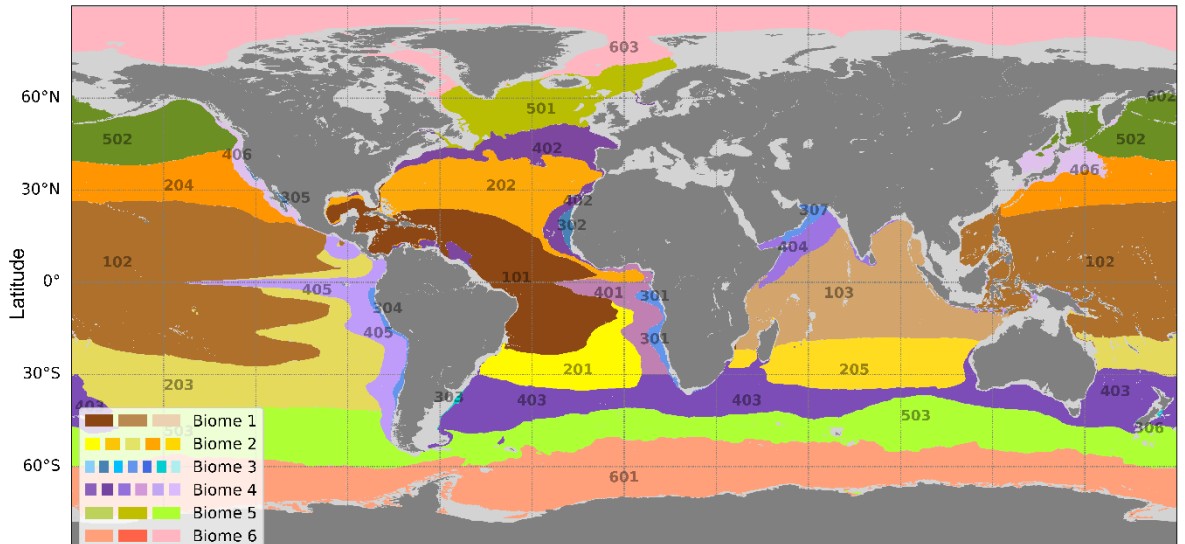

**Figure 1: Map of reference biophysical biomes obtained by PCA principal component clustering from averaged
epipelagic layer temperature, stratification, and NPP over the 1998-2023 time period. Geographical separation between
different areas of the same biome defines 27 associated provinces. Provinces are identified by different shades of biomes'
colors, defined in the legend. One label is attributed to each province with the hundreds' digits corresponding to the
biome in which they belong. Grey areas delimitate the domain where the depth of the water column is not sufficient to
ensure the existence of the three pelagic layers of SEAPODYM-LMTL (product ref 01, Table 1).**

The six reference biophysical biomes are characterized as: tropical, subtropical, eastern boundary coastal
upwelling systems, oceanic mesotrophic systems, sub-polar and polar (respectively numbered from 1 to 6). The
sub-division of these biomes according to ocean basin and hemisphere leads to the definition of 27 biophysical
provinces (identified by different shades of the biomes' colors in Figure 1).
The monthly time series of these provinces is available as an animation showing the provinces' geographical
evolution in time from 1998 to 2023 (https://doi.org/10.5446/68853). Together with the variations of ocean
environmental conditions, the geographical extent of provinces evolves in time.
The different biomes, and associated provinces, are characterized by specific environmental regimes (Figure 2).
Focusing on the biophysical conditions for each province, we consider the data distribution for averaged epipelagic
layer temperature, stratification, and NPP. Figure 2 shows monthly values of these three variables from 1998 to
2023, spatially averaged for each biome.

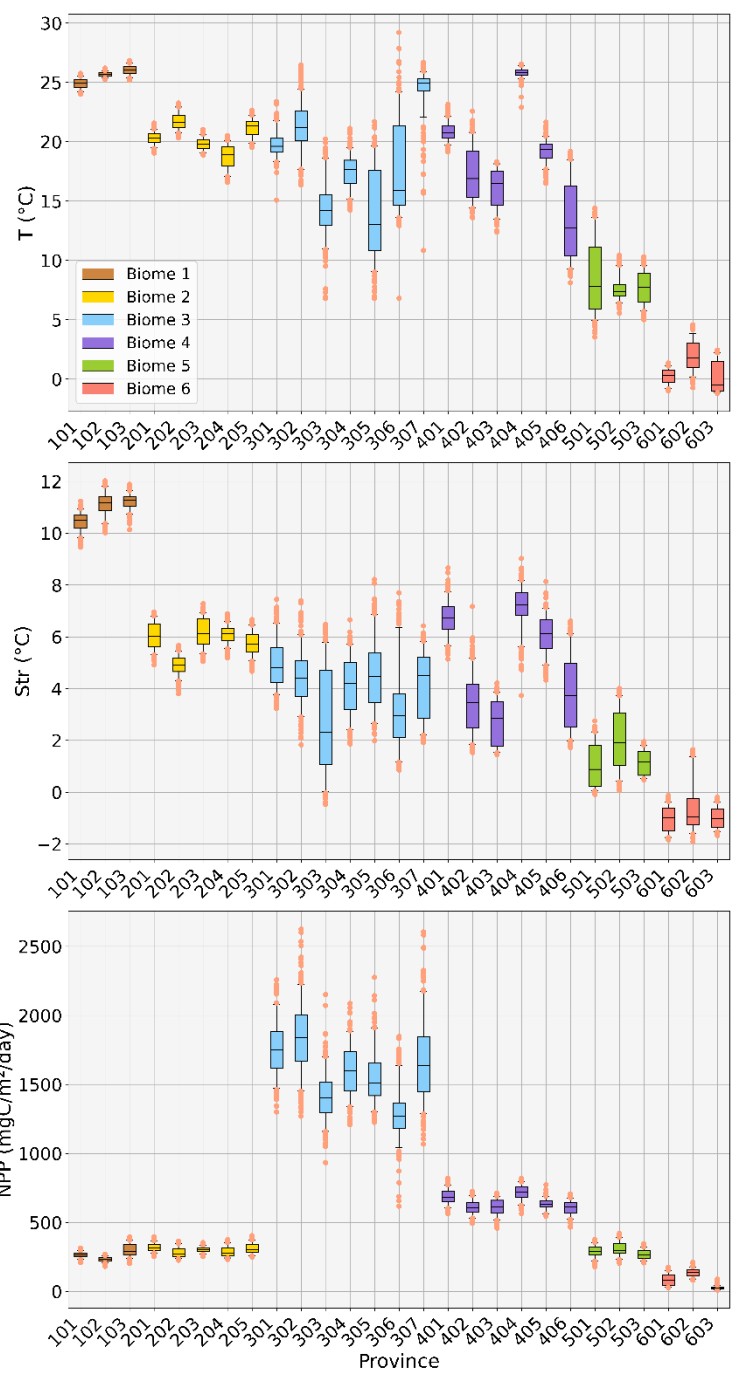

**Figure 2: Characterization of biophysical biomes using monthly environmental forcings: temperature of the epipelagic layer (T, °C), stratification (Str, °C), and NPP (mgC/m²/day) from 1998 to 2023. The analysis uses monthly values of T1, Str, and NPP, spatially averaged across each biome (i.e., one value per month for each environmental variable per biome). The boxplots depict the data distribution, with the median shown at the center of each rectangle, the first and third quartiles represented by the top and bottom edges of the rectangles, the whiskers extending to the 5th and 95th percentiles, and orange dots indicating outliers.**

Biome 1 (the tropical biome) is characterized by the warmest and most stratified waters, associated with relatively low biological production. A similar but less pronounced pattern is observed for Biome 2 (the subtropical biome). Biome 3 (the eastern boundary coastal upwelling systems) is by far the most productive biome. Biome 4 (the oceanic mesotrophic systems) also exhibits high NPP values, though weaker than Biome 3. Biome 5 (the sub-polar biome) is weakly stratified, characterized by cold waters, and shares a similar NPP range with Biomes 1 and 2. Biome 6 (the polar biome) features the weakest stratification and the lowest epipelagic layer temperatures among all biomes.

**3.2 Provinces' surface area evolution**

We aim to observe the geographical evolution of the provinces in time from 1998 to 2023. We provide in the Supplementary material, for each province, a scatter plot for the annual surface area for the period 1998-2023, with the associated linear regression (Figure S2). The slopes of the linear regression models computed from the annual time series of surface area for each province are computed (See supplementary material, Table S3, third column). These trends (in $km^2$/year) are also expressed as the equivalent percentage of evolution between 1998 and 2023, in % (Table S3, fourth column). The latter is displayed in Figure 3, as a map of the reference biophysical provinces showing their surface evolution in time from 1998 to 2023.

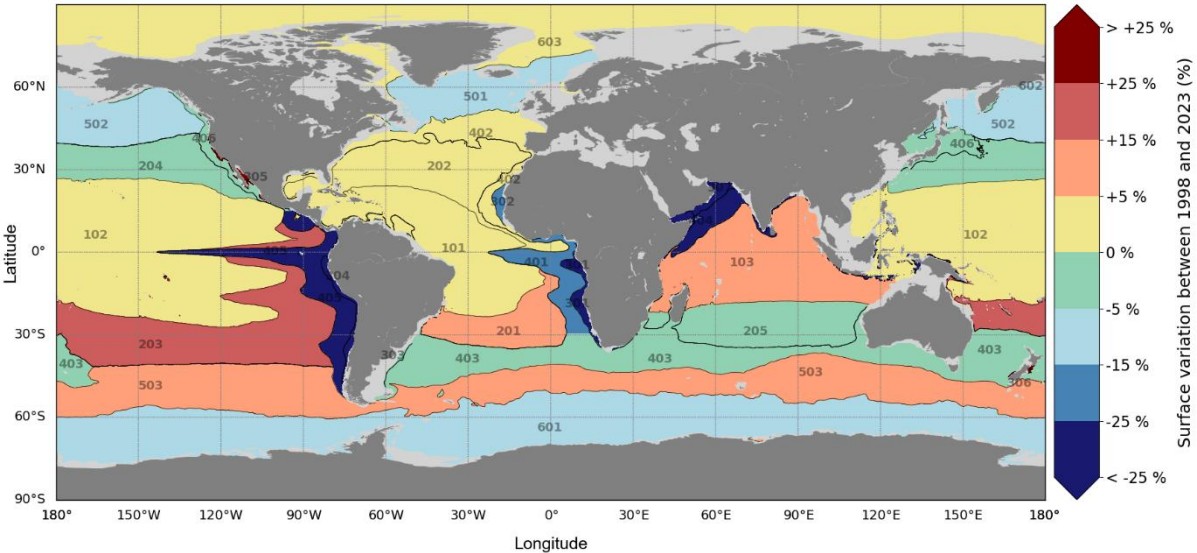

**Figure 3: Map of the provinces' surface area evolution in time from 1998 to 2023. Black lines delineate the definition of the 27 reference biophysical provinces (cf. Figure 1). Colors represent the trend in surface variation for each province (in % from 1998 to 2023): shades of red indicate increasing surface area, while shades of blue indicate decreasing surface area.**

From 1998 to 2023, there has been a decline in the surface area of productive provinces (i.e., characterized by high NPP) in eastern boundary coastal upwelling systems and oceanic mesotrophic systems (provinces belonging to Biomes 3 and 4, i.e. labelled 300's and 400's), as indicated by the provinces colored with shades of blue in Figure 3. Most of the polar and subpolar provinces such as the North Atlantic and North Pacific subpolar areas (respectively provinces 501 and 502) and the circumpolar province of the Southern Ocean (601) also display decreasing trends in their extent. On the other hand, provinces with increasing surface trends are mostly tropical or subtropical areas (Indian Ocean, South Atlantic tropical band, or South Pacific tropical band, respectively provinces 103, 201 and 203).

On a global scale, productive provinces and polar provinces seem to shrink in favor of tropical provinces expansion. However, some biomes exhibit significant discrepancies among the provinces they encompass. For instance, the surface of the Southern Ocean province 503 (belonging to the subpolar Biome 5) shows an increasing trend, in opposition with provinces 501 and 502 belonging to the same biome, showing decreasing trends in the northern hemisphere.

**3.3 Provinces' average latitude evolution**

Together with the evolution of provinces' surface area, provinces' average latitude is a valuable metric to track the geographical evolution of the provinces in time from 1998 to 2023. We provide in the Supplementary material, for each province, a scatter plot for the annual province's average latitude for the period 1998-2023, with the associated linear regression (Figure S4). The slopes of the linear regression models computed from the annual time series of average latitude for each province are computed (See supplementary material, Table S5). The poleward displacement of each province between 1998 and 2023 is displayed in Figure 4 (in degree poleward).

270

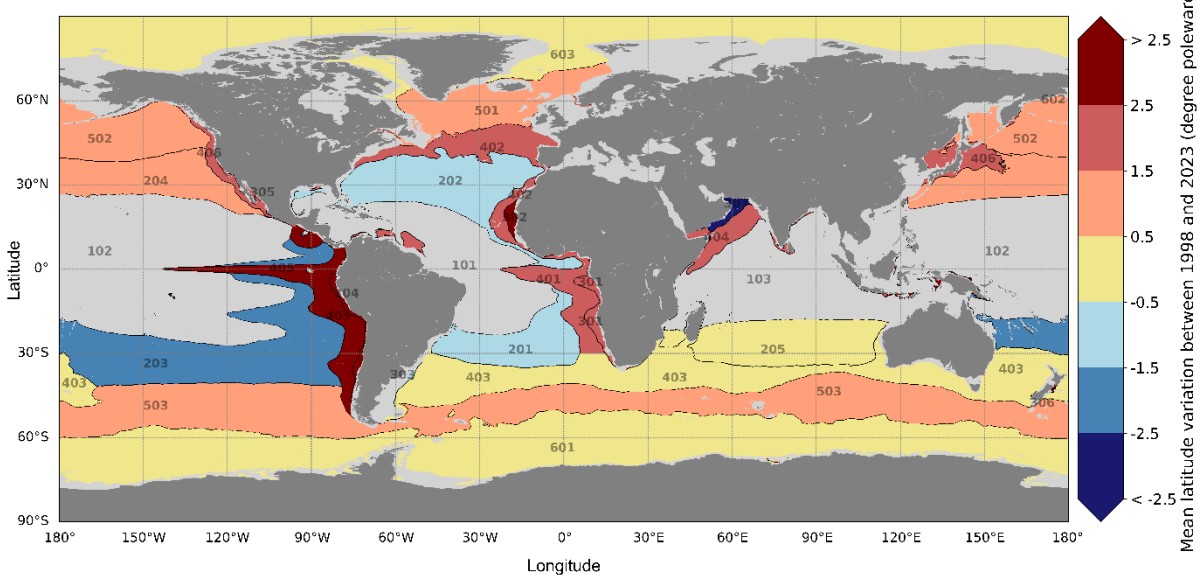

271

**Figure 4: Map of the provinces' average latitude evolution in time from 1998 to 2023. Black lines display the definition of the 27 reference biophysical provinces (cf. Figure 1). Colors represent the trend in average latitude variation for each province (in degree poleward from 1998 to 2023): darker shades of red indicate poleward drifting, while darker shades of blue indicate equatorward drifting. Provinces of the equatorial biome colored in grey (101, 102 and 103) are not considered in this diagnostic because of their equatorial position.**

Most of the provinces experience poleward drifting (provinces colored with shades of red in Figure 4). The tropical provinces displaying increasing surface trends (See provinces 201, 202, 203, Figure 3) experience equatorward drifting, as indicated by provinces colored with shades of blue in Figure 4. Provinces with average latitude evolution trends between +0.5 and -0.5 degree poleward over the time period are considered as stable in time, in terms of latitude (provinces colored in yellow in Figure 4. This range encompasses the 20% of provinces, exhibiting the least latitudinal drift over time, distinguished from the ones undergoing more pronounced and meaningful drifts.

**3.4 Sensitivity Analysis**

The robustness of the biophysical clustering obtained with the reference dataset, i.e., GLORYS12V1 for the physical variables and VGPM for the biological variable (see section 2.1., and table 1, product ref01), is tested by computing other biophysical clusterings derived from alternative environmental datasets. These alternative datasets include physical data from ARMOR3D (Guinehut et al., 2012; Mulet et al., 2012) and biological data from the biogeochemical model PISCES (Aumont et al., 2015),

The *Multi Observation Global Ocean 3D Temperature Salinity Height Geostrophic Current and MLD* product of Copernicus Marine Service (MULTIOBS_GLO_PHY_TSUV_3D_MYNRT_015_012, product ref03, table 1) provides 3-D temperature from ARMOR3D dataset, derived from an optimal analysis of 3-D observations. This product is used to compute the epipelagic layer temperature and the stratification, instead of GLORYS12V1 (used in reference biophysical clustering, product ref01, table 1). A first alternative clustering, employing the same methodology as the reference biophysical clustering (see 2.1. Variables and biophysical clustering), is performed using this product to compute the physical variables (the epipelagic layer temperature and the stratification), and still using VGPM (product ref01, table 1) to compute the NPP.

Then, the *Biogeochemical hindcast for global ocea*n product of Copernicus Marine Service (GLOBAL_MULTIYEAR_BGC_001_029, product ref02, table 1), is used to compute the NPP variable for the clustering instead of VGPM (product ref01, table 1). It provides 3D biogeochemical fields using PISCES biogeochemical model outputs. A second alternative clustering, employing the same methodology as the reference biophysical clustering, is performed using this product to compute the NPP, and still using GLORYS12V1 (product ref01, table 1) to compute the physical variables (as in the reference biophysical clustering).

The two alternative products mentioned above are available at ¼ degree from 1998 to 2022 at a monthly resolution.
Each of them is used to compute an alternative clustering (respectively using VGPM-ARMOR3D and PISCES-
GLORYS12V1). We compare our reference biophysical clustering (VGPM-GLORYS12V1, see Section 2.1),
downscaled from 1/12-degree to 1/4-degree resolution, with two alternative clusterings (Figure 5), all averaged
over the period 1998–2022.

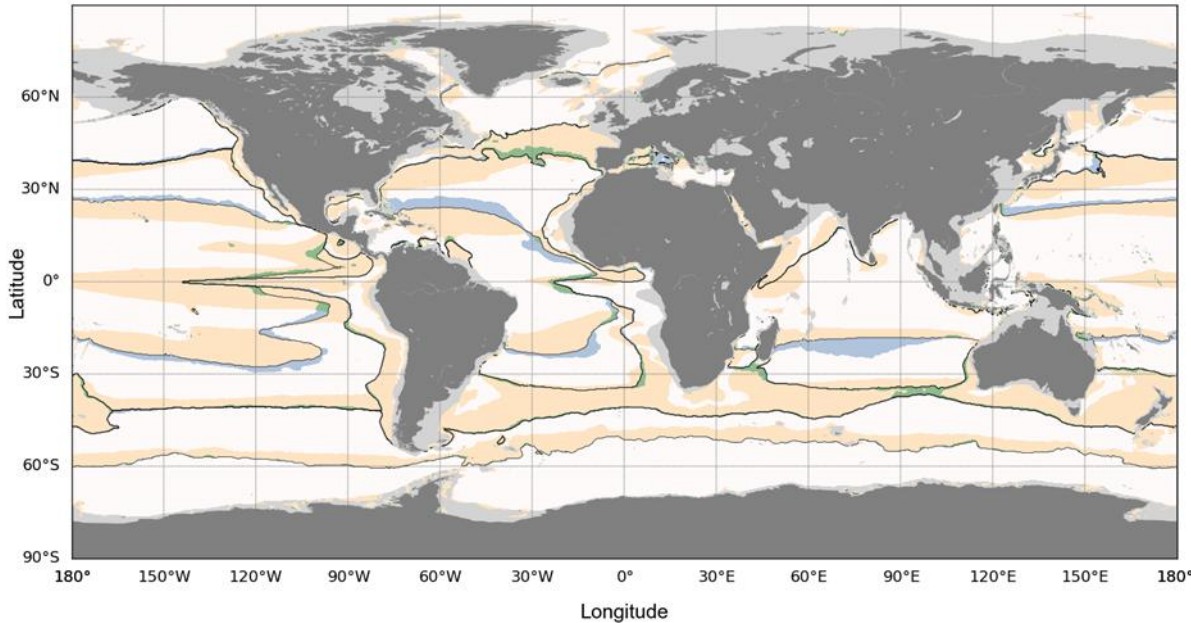


**Figure 5: Clustering sensitivity analysis. Map of reference biophysical biomes computed from GLORYS12V1 (product**
**ref01, table 1) and VGPM (product ref01, table 1), in black lines (cf. Figure 1). The white areas indicate where both**
**alternative clusterings assign the same cluster as the reference biophysical clustering. The blue areas indicate where the**
**clustering using ARMOR3D (product ref03, table 1) instead of GLORYS12V1 (product ref01, table 1) assign a different**
**cluster from the reference biophysical clustering. The yellow areas indicate where the clustering using PISCES (product**
**ref02, table 1) instead of VGPM (product ref01, table 1) assign a different cluster from the reference biophysical**
**clustering. The green areas indicate where both alternative clusterings assign a different cluster from the reference**
**biophysical clustering.**
Figure 5 shows that the clustering is very stable when changing the physical variable source from GLORYS12V1
(product ref01, table 1) to ARMOR3D (product ref03, table 1), as blue areas highlight minor boundaries
differences. However, when changing biogeochemical variable source from VGPM (product ref01, table 1) to
PISCES (product ref02, table 1), the productive biome 4 is highly impacted. However, NPP estimations from
PISCES (product ref02, table 1) and VGPM (product ref01, table 1) differ significatively. We notice that the
clustering remains relatively stable with respect to the source of forcings, although variations can arise when
forcing fields differ widely. The time series and results presented in the study are thus valid using VGPM and
GLORYS12V1 (both product ref01, table 1), but caution should be taken in extrapolating those results to clusters
issued from other biogeochemical sources (e.g. models' outputs).

**4 Discussion and conclusion**
In this study, we defined 27 biophysical provinces linked to micronekton, based on a methodology introduced in
Albernhe et al. (2024). Our definition of the reference biophysical provinces (Figure 1) has been compared to other
studies (e.g. Proud et al., 2017; Sutton et al., 2017, Ariza et al. 2022) employing comparable methodologies using
environmental variables to derive biogeographic regions. Sutton et al. (2017) classified regions based on
environmental drivers and expert knowledge, Proud et al. (2017) used clustering on environmental variables to
model deep scattering layers characteristics, and Ariza et al. (2022) derived provinces by clustering acoustic data
reconstructed from biophysical variables. Despite methodological differences among these three studies, the
resulting biogeographical regions closely align with ours. Notably, beyond the evident latitudinal banding (in the
austral Ocean for instance), more complex regional structures emerge in the North Atlantic, midlatitude frontal

zones (except in the South Pacific), and upwelling regions (see Figure 4 in Sutton et al., 2017; Figure 3A in Proud et al., 2017; Figure 2a in Ariza et al., 2022; and our Figure 1). This similarity likely arises because all approaches rely on biophysical variables that capture key information on temperature, biological productivity, and water column mixing.

The annual time series of biophysical provinces from 1998 to 2023 allows for tracking their temporal evolution capturing decadal to climatic variability, focusing on the variations in surface area and average latitude of each province. The resulting changes observed include a shrinking of productive and polar provinces, an expansion of equatorial and tropical provinces, and a poleward drift affecting most provinces.

Despite the complexity of the multifactorial causes behind the spatial variation of the provinces, we can attempt to infer the main environmental variables driving the provinces' spatial evolution and how changes in these variables contribute to their size variation and latitude drifting. Based on Figure 2, boxplots with median values particularly different from the others (e.g., either the highest or lowest) and narrower ranges (indicating low data variance) highlight the environmental variable most likely to characterize a province's specificity. Equatorial provinces (Biome 1) are characterized by very high temperature and stratification values, with narrow boxplots indicating weak data variance. Thus, temperature and stratification seem to be the most explanatory variables for these provinces. A similar but less pronounced pattern is observed in subtropical provinces (Biome 2). Productive provinces (Biomes 3 and 4) are highly distinguishable by their significantly elevated NPP values compared to others, suggesting that NPP is the most explanatory variable for these regions. Subpolar and polar provinces (Biomes 5 and 6, respectively) are marked by low stratification and cold waters, with polar provinces showing the weakest values among all biomes. Therefore, temperature and stratification appear to be the key explanatory variables for subpolar and polar provinces.

The spatial variation of provinces is a multifactorial outcome of expected environmental change in time. This variation arises from complex interactions and feedback mechanisms among biophysical variables, driven by a range of intricate physico-biogeochemical processes. Under global warming, the ocean is warming (Abraham et al., 2013, Kwiatkowski et al., 2020, IPCC 2021), which is consistent with the increasing trend observed in the time series of global mean epipelagic layer temperature (Figure S1). The ocean density structure and vertical dynamics also change (Srokosz and Bryden, 2015), which aligns with the increasing trend observed in the time series of global mean stratification of the mesopelagic ocean (Figure S1). Both primary production and vertical displacements of phytoplankton are impacted by these physical changes (Denman and Gargett, 1983; Laufkötter et al., 2013), including stratification leading to nutrient limitation. Phytoplankton growth is deeply influenced by temperature (Grimaud et al., 2017), but many other features such as nutrients supply or light induce variability in NPP patterns (Behrenfeld et al., 2006). A global decrease in NPP has been observed in early 21$^{st}$ century (Behrenfeld et al., 2006; C. Laufkötter et al., 2013; Kwiatkowski et al. 2020), which is consistent with the decreasing trend observed in the time series of global mean NPP (Figure S1). However, an analysis of remotely sensed surface chlorophyll-*a* concentration (upon which NPP calculations are based) reveals highly contrasted trends between available merged products (Pauthenet et al. 2024), questioning the accuracy of these results. Moreover, future climate model projections of global NPP over the 21st century display a poor level of confidence.

In the present study, we observe a global shrinking of productive provinces and polar provinces, in favor of an expansion of equatorial and tropical provinces (Figure 3). A regional decline in primary production could cause the reduction in the size of productive provinces, with NPP identified as the most probable explanatory variable for provinces within biomes 3 and 4. The previous reference to Pauthenet et al.'s work advises caution in interpreting this. Concurrently, the global increase in ocean temperature over the past decades explains both the expansion of equatorial and tropical provinces and the contraction of polar provinces, with temperature identified as the key driving factor for these provinces. Trends in ocean temperature, supported by converging estimations, are much more robust and pronounced than those of NPP (Bopp et al., 2013).

The latitudinal patterns of our provinces' definition are directly impacted by temperature changes. These latitudinal patterns identify the equatorial, subtropical, subpolar and polar biomes (Biomes 1, 2, 5 and 6) which were previously suggested to be primarily influenced by temperature and stratification variables. The increase in ocean temperatures drives the expansion of equatorial provinces, causing their climatic boundaries to shift poleward while they remain centered around the equator. A similar poleward drift is observed in subtropical provinces. Likewise, polar provinces are affected by warming, as the reduction in cold water areas confines them to higher latitudes and pushes their climatic boundaries poleward. Tracking the geographical evolution of these provinces

over time, as illustrated in Figure 4, demonstrates that most provinces exhibit this poleward drift, apparently likely driven by temperature.

In addition to the provinces' definition methodology, the previous publication upon which the present study is based (Albernhe et al., 2024) demonstrates that each province features a specific characterization in terms of micronekton biomass and vertical structure. Following the hypothesis that these characteristics are preserved over time, which needs to be further investigated, the evolution of provinces' surface area can account for global micronekton trends and estimations. For instance, the shrinking of provinces featuring the highest density of micronekton biomass would lead to a global decrease of micronekton biomass. Productive provinces and subpolar provinces are characterized by high densities of micronekton biomass (Albernhe et al., 2024), whereas equatorial and tropical ones display weaker densities of micronekton biomass. If provinces characteristics are preserved over time, the shrinking of productive and polar provinces together with the expansion of the equatorial and tropical provinces would imply a global decline of micronekton biomass. This reasoning is based on a basic deduction from the consequences of ecological niche surface variation. However, the underlying mechanisms that could explain a global decline in micronekton biomass may be partly attributed to the previously mentioned decreasing trend in NPP at the global scale. Since NPP is at the base of the trophic chain, this decline has cascading effects up to micronekton, limiting their energy sources and thus reducing population development. Additionally, the potential global decline in micronekton biomass may also be partly induced by the increasing trend in global ocean temperature, which affects micronekton development times (Gillooly et al., 2002), including growth and mortality.

This potential trend for micronekton biomass evolution on the historical period would be in range with studies on micronekton biomass climate projections (Bryndum-Buchholz et al., 2018; Kwiatkowski et al, 2019; Lotze et al, 2019; Tittensor et al., 2021; Ariza et al., 2022). In Ariza et al. (2022), the authors derive acoustic provinces from a clustering using acoustic data as a proxy of micronektonic biomasses, which they reconstructed from biophysical data (satellite-derived chlorophyll concentration, sea surface temperature and subsurface dissolved oxygen). In Albernhe et al. (2024), our biophysical provinces were compared to these acoustic provinces, revealing a strong overall agreement, particularly in terms of latitudinal patterns, dynamic regions, and upwelling areas. However, our clustering method did not capture oxygen-driven patterns. Ariza et al. (2022) also explored the spatio-temporal variability of provinces in future projections extending to 2100. They predict a contraction of upwelling and subpolar provinces alongside an expansion of subtropical and temperate provinces, leading to a global decline of pelagic fauna. Despite the differing timeframes of the two studies (1998–2023 in our case vs. projections for 2000–2020 and 2080–2100 in Ariza et al.), their conclusions align with ours regarding the trends observed in these biogeographical provinces, and the consequences on mid trophic biomass. To end up regarding the poleward drifting of provinces, this valuable observation in range with the literature (Hastings et al., 2020; Pinsky et al., 2020), suggesting a potential poleward migration of micronektonic populations induced by temperature changes.

The authors would, however, like to draw the reader's attention to the caution required when interpreting the trends. First, the trends presented in this study (whether regarding the surface area of provinces or their mean latitude) are quantified based on linear regressions. However, these regressions exhibit low $R^2$ values, indicating that the linear relationships are not statistically significant. A 26-year period is insufficient to establish statistically robust linear trends in the characteristics of the provinces under investigation. Moreover, observed trends can vary depending on the temporal scale of the study: short-term trends under 26 years do not necessarily reflect long-term, sustainable changes. Additionally, the data employed in this analysis are subject to various biases and uncertainties (e.g., discrepancies between products estimating chlorophyll-a concentration used to derive NPP, as noted in Pauthenet et al., 2024). Finally, a detailed analysis of the consistency of each province's micronektonic characteristics over time (regarding biomass and the vertical structure of micronekton), should be conducted. This would provide a more solid basis for confirming the link between changes in province surface areas and the evolution of total micronekton biomass. Misinterpreting these findings could lead to premature conclusions or ineffective communication to the public, thereby increasing the risk of misinformation about critical issues such as climate change.

Despite these uncertainties, the indicators defined in this study show sensitivity to changes in environmental parameters and are valuable metrics that should be monitored over the long term. Examining these parameters over longer timescales may allow us to identify climate trends, with the significance of these trends increasing as the time series extends.
















**Video supplement**

Sarah, Albernhe; Thomas, Gorgues; Olivier, Titaud; Patrick, Lehodey; Christophe, Menkes; Anna, Conchon: Biophysical provinces, monthly times series 1998-2023. Copernicus Publications, 2024. https://doi.org/10.5446/68853

**Data availability**

All data products used in this paper are listed in Table 1, along with their corresponding documentation and online availability.

**Table 1: Product Table**

| Product ref. No. | Product ID & type | Data access | Documentation |
|---|---|---|---|
| 01 | GLOBAL_MULTIYEAR_BGC_001_033; Numerical Models | EU Copernicus Marine Service Product: *Global Ocean low and mid trophic levels biomass content hindcast*, Mercator Ocean International, *https://doi.org/10.48670/moi-00020* | Quality Information Document (QUID): Titaud et al., 2023<br><br>Product User Manual (PUM): Titaud et al., 2023 |
| 02 | GLOBAL_MULTIYEAR_BGC_001_029; models | EU Copernicus Marine Service Product: *Biogeochemical hindcast for global ocea*n, Mercator Ocean International, https://doi.org/10.48670/moi-00019 | Quality Information Document (QUID): Perruche et al., 2019 |

| | | | Product User Manual (PUM) : Le Galloudec et al., 2022 |
|---|---|---|---|
| 03 | MULTIOBS_GLO_PHY_TSUV_3D_MYNRT_015_012; In-situ observations, Satellite observations | EU Copernicus Marine Service Product: *Multi Observation Global Ocean 3D Temperature Salinity Height Geostrophic Current and MLD*, CLS, https://doi.org/10.48670/moi-00052 | Quality Information Document (QUID): Greiner., 2023 Product User Manual (PUM) : Verbrugge et al., 2023 |

465

## Author contributions

SA produced the clustering, the provinces' timeseries, and the associated diagnostic for their evolution in time. SA wrote and edited the report.

TG was involved in the investigation process as an advisor and reviewed the report.

OT was involved in the investigation process as an advisor and produced the lower and mid-trophic levels biomass density (*Global Ocean low and mid trophic levels biomass content hindcast*, GLOBAL_MULTIYEAR_BGC_001_033) 1998-2023 time-series.

The other authors are supervising the PhD of SA, and were involved in the investigation process to define the methodology used for the clustering (Albernhe et al., 2024)

## Competing interests

The contact author has declared that none of the authors has any competing interests.

## Disclaimer

## Acknowledgements

This work was funded by the NECCTON project, which has received funding from Horizon Europe RIA under grant agreement No 101081273. The views, opinions and practices used to produce this study are however those of the author(s) only and do not necessarily reflect those of the European Union or European Research Executive Agency. Neither the European Union nor the granting authority can be held responsible for them.

## Financial support

This work was partly funded by the GLO-RAN component of the Copernicus Marine Environment Monitoring Service (21003L04-COP-GLO RAN-4300) and the Horizon Europe RIA (Grant Number 101081273) & UK Research and Innovation NECCTON project. PL 's contribution was funded by the European Union under grant agreement no. 101083922 (OceanICU).

## Review statement

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
