# Peer review of "Micronekton indicators evolution based on biophysically defined provinces"

_State of the Planet, 2024_

## Referee Comment (RC2)

**Review**

The paper titled "Micronekton Indicators Evolution Based on Biophysically Defined Provinces" by Albernhe et al. examines the evolution of micronekton indicators across biophysically defined provinces over a 25-year period, from 1998 to 2023.

This study offers a dynamic, long-term analysis of micronekton provinces, providing quantitative insights into surface area and latitude trends while presenting direct evidence of climate-driven shifts in marine ecosystems. It builds upon previous research by offering a more precise, time-resolved, and geographically validated understanding of micronekton variability.

By focusing on two key indicators—surface area trends and average latitude shifts—this study highlights metrics that could serve as future Ocean Monitoring Indicators. Its straightforward and impactful contribution makes it highly relevant for the OSR.

My primary concern is that this paper heavily relies on the work by Albernhe et al. (2024), which is still under review and not yet publicly available. Additionally, no reference to this paper is provided, leaving significant gaps in the current contribution. Without the publication of Albernhe et al. (2024), too many critical questions remain unanswered.

Therefore, unless Albernhe et al. (2024, under review) is officially published, I must recommend rejecting this submission. However, given the relatively long publication process for OSR9, this contribution could be accepted on the condition that the referenced paper is published before the final release of OSR9.

Furthermore, I suggest revisions, and the authors should provide a copy of Albernhe et al. (2024, under review) alongside their revisions to allow for a thorough evaluation of the current submission.

Major comments

1) It remains completely unclear how the 27 provinces are obtained. I suggest that authors will provide more information about the determination of 27 provinces in Materials and Methods section. Otherwise, the paper cannot be read as stand alone paper.

2) The manuscript will benefit if more in-depth discussion in comparison to the works by (Reygondeau et al., 2012), (Sutton et al., 2017), (Costello et al., 2017), (Elizondo et al., 2021), (Proud et al., 2017), (Ariza et al., 2022) is provided.

Detailed comments

L172-174 The following statement is too general "Seasonal variability can be observed with the latitudinal shifts of the horizontal boundaries, as well as regional seasonal phenomena or isolated phenomena like ENSO events." Without any hints on what to look from the video, it requires detailed analysis of the video itself by readers.

Section 3.4 More appropriate title is Sensitivity analysis instead of Uncertainty. Text of the section should be adjusted accordingly.

---

## Author Comment (AC1)

**Response to Referee #1 comments**

We sincerely appreciate the reviewer's constructive comments and valuable suggestions, which have contributed to enhancing the clarity and robustness of our manuscript. In this document, the reviewer's comments are presented in black, while our associated responses are presented in blue. Additionally, we provide the revised version of the manuscript.

**Overview and general recommendation:**

This manuscript examines the variation in surface area and latitude of ocean provinces, which are defined by three environmental variables closely linked to micronekton biomass. The topic is very interesting and valuable for marine and climate change policy-making. However, issues related to clarity, methodological transparency, and result interpretation significantly undermine the scientific rigor and credibility of the study. Therefore, I recommend a major revision.

**Specific comments:**

1. A major concern is that the entire study extends from an unpublished paper currently "under review," as mentioned over ten times throughout the manuscript. However, the authors provide no additional materials to clarify the methodology or findings of the prior study. Without access to this prior work, the present study lacks context and transparency. For example, the methodology is frequently referenced as "see the previous unpublished study", but the reader know nothing about that. I attempted to locate the referenced study (Albernhe et al., 2024) online to find whether it is published so I will understand what authors talking about. Fortunately, it is published. If this study heavily depends on the unpublished paper, I suggest waiting for its publication or providing supplemental materials to ensure reviewers and readers fully understand the background and methodology.

   ➔ Thank you for your thoughtful feedback. We appreciate your concern regarding the reference to the unpublished paper and the importance of providing sufficient context for our study. We are pleased to inform you that the referenced study, "Albernhe et al., 2024," has since been accepted and published. The full reference is as follows:

   *Albernhe, S., Gorgues, T., Lehodey, P., Menkes, C., Titaud, O., De La Giclais, S. M., & Conchon, A. (2024). Global characterization of modelled micronekton in biophysically defined provinces. Progress in Oceanography, 229, 103370.*

   We have updated the manuscript to reflect the status of this publication (i.e., Introduction l73, l93, Material and methods l100, l103, l116, etc.), and added the reference in the appropriate section.

   However, we want this paper to be comprehensive on its own. The different steps of the methodology are detailed in Section 2.1., l102 : *"While the overall methodology is detailed in Albernhe et al. (2024), we outline the different steps of the method below to ensure this study is comprehensive and self-contained. [...]"*

2. Lines 56-59: The authors mention that micronekton observations primarily rely on trawl sampling, which can introduce biases, and then abruptly shifts to ecosystem and population models. The connection is unclear. Why are numerical models, such as spatial ecosystem and population models, complementary tools? Please elaborate and provide supporting details.

   ➔ We have clarified the text to better explain why numerical models are complementary to field observations. Micronekton observations mainly rely on (i) ship-borne acoustics and (ii) trawl sampling, both of which have significant limitations. Trawl sampling can introduce

biases due for instance to species avoidance and provides only coarse, localized data. Acoustic measurements do not yet offer a reliable estimation of micronekton biomass. We consider numerical models as complementary tools as they overcome these biases and offer a continuous representation of biomass across space and time.

➔ Introduction, l53 : *"Therefore, estimating micronekton biomass is a major concern for fisheries management and climate regulation. Direct observations of micronekton primarily rely (i) on ship-borne acoustic measurements, which does not provide yet a reliable representation of the micronekton biomass (McGehee et al., 1998; Kloser et al., 2002), and (ii) on trawl sampling, which is susceptible to biases due for example to species avoidance (Kaartvedt et al., 2012) and has a coarse sampling. Numerical models, such as the Spatial Ecosystem and Population Dynamics Model – Low and Mid Trophic Levels (SEAPODYM-LMTL: Lehodey et al., 2010; 2015; Conchon, 2016) are complementary tools for studying micronekton biomass. Indeed, by simulating micronekton dynamics based on key biological and physical processes (such as growth, recruitment, mortality and environmental influences), these models provide a continuous representation of micronekton biomass across space and time. This helps fill observational gaps, enabling the analysis of large-scale patterns, the simulation of future scenarios, and ultimately a better understanding of the mesopelagic ecosystem."*

3. Lines 63-64: The authors state that "catch per unit of effort of commercial fisheries" and "multi-expertise discussions" are environmental forcings. However, these appear to be more related to human or economic activities. Please clarify or revise.

➔ Thank you for your comment. We agree that "catch per unit of effort of commercial fisheries" is not an environmental forcing, and we have revised the sentence to clarify this point. Additionally, we would like to clarify that the "multi-expertise discussions" refer to a focus on biogeographical features. Specifically, as outlined in Sutton et al. (2017), the panel involved expertise in a variety of disciplines, including descriptive and numerical physical oceanography, geospatial mapping, marine ecology, organismal biology, and deep-pelagic taxonomy. Since the term "environmental forcing" is not directly applicable, we have rephrased the sentence accordingly.

➔ Rephrased paragraph, l66 : *"Various combinations of features have been used to create accurate definitions of provinces for each field: environmental features such as the distribution of species (Costello et al., 2017) and phytoplankton species assemblages (Elizondo et al., 2021), biogeographic insights from multi-expertise discussions (Sutton et al., 2017), and fisheries-related data, such as catch per unit of effort of commercial fisheries (Reygondeau et al., 2012)."*

4. Lines 70-84: This paragraph is confusing. The statement, "The ambition of this work was to identify..." seems to describe the prior study rather than the present one. After multiple readings, it appears the authors are referencing the previous work. I suggest revising this section to explicitly distinguish between the goals of the prior study and the current one.

➔ We have rephrased this paragraph, referring to Albernhe et al. (2024) as '*the prior study*' and the present study as '*the present study*', and emphasizing the distinction between the two studies.

➔ Rephrased penultimate paragraph, l74: "[...] *Since the present study builds upon Albernhe et al. (2024), we detail the main and key findings of the prior study in the following sentences. The ambition of the prior study Albernhe et al. (2024) was to [...]. "*

➔ Rephrased ultimate paragraph, l91: "*In the present study, we focus on provinces' features, such as surface area and positional changes, which serve as valuable indicators providing insights into the evolution of ecosystem structure over time, both globally and*

*regionally. Following Albernhe et al. (2024)'s methodology, we define in the present study [...]"*

5. Lines 114-116: I understand that authors used the same methodology as the prior study. However, the main difference is the period of input data (1998-2019 vs. 1998-2023). Despite conducting a new training phase with updated data, the authors still used k=6 for clustering, as in the prior study. It is necessary to show the results of clustering and PCA in the Result part or Supplementary materials.

➔ The new PCA yields results highly consistent with those from Albernhe et al. (2024). The first two principal components account for 98.1% of the variance (compared to 97.9% in Albernhe et al., 2024) and are expressed as follows:

$$F1 = 0{,}612\ T + 0{,}563\ Str + 0{,}556\ NPP$$

$$F2 = -0{,}306\ T - 0{,}480\ Str + 0{,}822\ NPP$$

*(F1 = 0,659 T + 0,620 Str + 0,427 NPP; F2 = -0,174 T - 0,420 Str + 0,888 NPP in Albernhe et al. 2024)*

➔ The results of the PCA have been detailed in the Result section (Section 2.1., l121) : *"As described in Albernhe et al. (2024), a Principal Component Analysis (PCA) (Hotelling, 1933) is performed on the three environmental variables mentioned above (i.e., epipelagic layer temperature, stratification and NPP), producing empirical orthogonal functions that strongly mirror those identified in Albernhe et al. (2024). We selected the two principal components that explain the most variance, accounting for 98,1% of the variance (68,2% and 29,9% for the first and second PCA respectively)."*

➔ In our previous study (Albernhe et al. 2024), we were initially unaware of the number of relevant biophysical biomes or the magnitude of how global-scale data for the three environmental variables could be classified. The determination of six clusters (k = 6) was a result of this exploration, accurately identifying the number of clusters needed. In the current study, we aim to replicate the same methodology, including the use of six clusters. Recomputing metrics identifying the optimal number of clusters with the new training dataset would yield the following plots: silhouette metric (left) and elbow metric (right).

[Figure]

[Figure]

The silhouette metric results but suggests an optimal number of clusters at k = 5 (peak/local maximum), which is slightly more distinct than the others. Meanwhile, the elbow metric does not indicate any optimal number of clusters being significantly more accurate than others (a sharp slope break forming an elbow). To remain consistent with the previous study (Albernhe et al., 2024), we retain k = 6 clusters, very close to the k = 5 suggested by the silhouette metric.

Although k = 5 could have been the optimum solution, k = 6 is still a valid solution (also because the difference in the time period does not change our empirical orthogonal functions) and it enables the use of our prior findings, where we characterized the six biomes (and 27 derived provinces) in terms of modeled micronekton biomass and vertical structure. The conclusions drawn highlighted the effectiveness of these six biomes in identifying regions with demonstrated accuracy in homogeneous micronekton characteristics. Consequently, our objective here is to retrieve a similar classification of global ocean, to build further analysis upon the previous study outcomes.

➔ Section 2.1., l 131 : *"In Albernhe et al. (2024), we identified six clusters (k = 6) to classify global-scale environmental data, effectively distinguishing biophysical biomes. In this study, the different metrics used to determine the optimal number of clusters do not exhibit a strongly pronounced pattern. One suggests that k = 5 could be a suitable choice, albeit not with strong certainty. To ensure consistency with Albernhe et al. (2024), we maintain k = 6, allowing us to build upon our previous findings on micronekton biomass and vertical structure."*

6. Line 122: Authors mentioned they are using monthly time series of biophysical biomes that captures seasonal and interannual variability, but I cannot see any detailed description in the whole manuscript and Supplementary about that. Also, this is my another main concerned of the study. Please see the comment point 7.

➔ The initial data used in this study are indeed available at a monthly resolution. To adhere to the methodology outlined by Albernhe et al. (2024), monthly provinces are derived from the clustering. While the provinces are available at a monthly resolution, our analysis focuses on interannual to climatic trends and therefore using annual provinces. The aim of the OSR is to report the evolution of marine environment of the global ocean over the past decades up to close to real time. Following this purpose, we concentrate this study on interannual to climatic trends in the surface area and mean latitude of the provinces to identify long-term patterns in these metrics. The complexity of seasonal variations is beyond the scope of this work and would require a separate, detailed investigation.

➔ Reformulation and clarification, Section 2.2, first paragraph, l152 : *"The aim of this study is to analyze the evolution of the provinces in time from 1998 to 2023. The biophysical data described in the previous section are available at a monthly resolution and monthly provinces are derived through clustering, in order to follow Albernhe et al. (2024)'s methodology. While provinces are resolved monthly, our analysis focuses on decadal to climatic trends aiming to identify long-term patterns in their evolution. To study the temporal variability and identify potential trends over the 26 years, we consider the annual time series. We calculate indicators based on the monthly definition of provinces and then compute the annual averages of these indicators. "*

7. Lines 136-150: The authors used linear regression ($R^2$ and slope) to analyze trends in surface area and poleward shifts, but this approach raises several issues:

First, there is insufficient information on how surface area and poleward shifts vary annually. Scatterplots showing surface area and average latitude trends, with fitted regression lines, should be included in the Results or Supplementary Materials.

➔ We acknowledge that these plots should be included. As there are 27 provinces, and thus 27 plots, and two different diagnostics, we included them in the Supplementary material (Figure S2, S4).

➔ Section 3.2., L237 : *"We provide in the Supplementary material, for each province, a scatter plot for the annual surface area for the period 1998-2023, with the associated linear regression (Figure S2)."*

➔ Section 3.3, L265 : *"We provide in the Supplementary material, for each province, a scatter plot for the annual province's average latitude for the period 1998-2023, with the associated linear regression (Figure S4)."*

Second, low $R^2$ values (approaching zero) suggest poor linear regression fits. Statistical significance of the regression should also be reported. Seasonal and interannual variations may not follow a simple linear relationship. The observed dynamics could involve regular or cyclical changes over time, which are not captured by linear regression.

➔ Indeed, the regressions show low $R^2$ values, as you correctly pointed out: the trends are not linear, at least the linear regression are not statistically significant. A 26-year period is too short to identify statistically significant trends on the provinces' features we focus on. The description of $R^2$ values in the paper in Section 2.2 could be misleading, as the aim of the analysis is to identify the direction and relative magnitudes of the trends, not to prove their statistically significant linearity. To ensure transparency with the reader, and as you suggested in your previous point, we have added scatter plots and the corresponding regressions for each province, for both metrics (surface and latitude).

➔ Thank you for your insightful comment highlighting the importance of caution and nuance in interpreting the results of this study. It is crucial to guide readers to avoid drawing oversimplified or misleading conclusions. Including the scatter plots in the supplementary materials will allow readers to directly observe the time series, appreciate the complexity of analyzing environmental metrics over short periods.

➔ Section 2.2 last paragraph, l186: *"Recapitulative tables for each of these two metrics are provided in the supplementary material (Table S3 for surface area and Table S5 for mean latitude). These tables present, for each province, the trend from the linear regression model, the total variation over the 26 years, and the coefficient of determination ($R^2$) for each regression. $R^2$ is a statistical measure that evaluates the degree of fit between the observed values and the linear regression model, allowing the statement of statistically significant linear trends. A 26-year period is too short to detect statistically significant trends in such biophysical features. Due to interannual variability, $R^2$ values are not expected to be close to 1, which would indicate statistical significance of the linear trends. The purpose of the linear regressions is to identify the direction and relative magnitude of the trends, rather than to confirm their statistical significance. Caution must be taken while considering such trends. Thus, scatter plots of the annual time series for surface area (Figure S2) and mean latitude (Figure S4), with the corresponding linear regression, are provided in the supplementary material for each province. These plots allow for direct observation of the time series."*

Third, Conclusions based on a comparison of 1998 and 2023 alone are unconvincing. Given the low $R^2$ values, comparing other years (e.g., 1998 vs. 2022 or 2021) could yield opposite conclusions for some provinces.

➔ Thank you for your valuable feedback. We acknowledge that the low $R^2$ values indicate a noticeable level of variability between the years compared, and we agree that interannual variability impacts the trends observed. We examine the time period extending to 2023, as requested for the Ocean State Report, but we acknowledge that comparing other years could lead to more nuanced conclusions. This variability emphasizes the importance of considering longer-term trends and the influence of interannual fluctuations when drawing conclusions (see our response to your next comment, and the last paragraph of the discussion Section).

➔ In terms of methodology, we consider on the slope of the linear regression, which indicates an annual change of x km²/year for the surface area diagnostic and an annual shift of y degrees/year (poleward or equatorward) for the mean latitude diagnostic. This approach, based on the linear regression computed over the entire time series, captures the trend across the full period from 1998 to 2023. Rather than comparing the first and last year of the time series, we want to project the equivalent evolution over 26 years based on the slope of the trend. Thus, we compute the slope of the linear regression (either in km²/year or degrees/year) multiplied by 26 years. This provides the equivalent 26-year trends in terms of (i) total surface area increase or decrease per province (in km²) and (ii) total mean latitude shift (in degrees). For (i), the surface area needs to be scaled by the surface area of the province at the beginning of the study (i.e., 1998), given the very diverse provinces' sizes.

➔ The terminology used in the paper may have been misleading, so we have revised it for clarity.

Section 2.2, second paragraph, l161: *"To evaluate the evolution of surface area over time, our approach is based on a simple linear regression model applied to the annual surface area (in km²) of each province from 1998 to 2023. We analyze the slope of the regression (in km²/year), to account for the direction and first-order magnitude of variability. Rather than directly comparing the years 1998 and 2023 to quantify the variation between these dates (which would assume that surface areas for these two years perfectly align with a statistically significant linear trend), we project the equivalent evolution over 26 years based on the slope of the regression (i.e., 26 × slope). From this projected variation, we compute the percentage change in surface area over 26 years (in %) relative to the surface area at the start of the time series (year 1998)."*

Section 2.2, third paragraph, l169: *"Similarly, to track the poleward drift of provinces over time, our approach is based on a simple linear regression model applied to the average latitude of each province of each province from 1998 to 2023. We analyze the slope of the regression (in degrees poleward/year), to account for the direction and first-order magnitude of variability. The 'degree poleward' unit that we use for this diagnostic is associated with degree N for provinces in the northern hemisphere, and degree S for provinces in the southern hemisphere. Thus, provinces belonging to the equatorial Biome 1 (provinces 101, 102 and 103) are not considered in this diagnostic because of their equatorial position. Rather than directly comparing the years 1998 and 2023 to quantify the variation between these dates, we project the equivalent evolution over 26 years based on the slope of the regression (in degree poleward), following the same approach as described for the surface area metric."*

So, I recommend using more sophisticated methods, such as Generalized Additive Models (GAMs), Multivariate Adaptive Regression Splines (MARS), Seasonal Autoregressive Integrated Moving Average (SARIMA), or machine learning approaches, to analyze seasonal and interannual variations. These methods could also help identify the significance of different environmental variables and their roles in driving variations in surface area and poleward shifts, enhancing the study's credibility and depth.

➔ We acknowledge that more advanced statistical or machine learning approaches could potentially uncover more complex patterns in the time series, providing additional insights into trends within our data. However, as outlined in previous responses, this study focuses on interannual to climatic trends in the surface area and mean latitude of the provinces to identify long-term general patterns in these metrics. While the authors agree on the complexity of seasonal variations and the lack of statistically significant linearity in the trends during the time series analysis, investigating these aspects in detail lies beyond

the scope of this work and would require a dedicated study. The slope of the regression analyzed here provides a straightforward assessment of the direction and first-order magnitude of the trends, offering a basic yet informative perspective on the temporal evolution of the provinces. However, including the scatter plots in the supplementary materials as you suggested give more insights on the time scales, strength, and significance of the variations in the time series.

→ We have added a paragraph at the end of the discussion section for transparency with the reader, emphasizing the various nuances that should be considered when interpreting the results of this study: *"The authors would, however, like to draw the reader's attention to the caution required when interpreting the trends. First, the trends presented in this study (whether regarding the surface area of provinces or their mean latitude) are quantified based on linear regressions. However, these regressions exhibit low $R^2$ values, indicating that the linear relationships are not statistically significant. A 26-year period is insufficient to establish statistically robust linear trends in the characteristics of the provinces under investigation. Moreover, observed trends can vary depending on the temporal scale of the study: short-term trends under 26 years do not necessarily reflect long-term, sustainable changes. Additionally, the data employed in this analysis are subject to various biases and uncertainties (e.g., discrepancies between products estimating chlorophyll-a concentration used to derive NPP, as noted in Pauthenet et al., 2024). [...] Misinterpreting these findings could lead to premature conclusions or ineffective communication to the public, thereby increasing the risk of misinformation about critical issues such as climate change.*

*Despite these uncertainties, the indicators defined in this study show sensitivity to changes in environmental parameters and are valuable metrics that should be monitored over the long term. Examining these parameters over longer timescales may allow us to identify climate trends, with the significance of these trends increasing as the time series extends."*

→ The relative roles of environmental variables in surface and latitude variations have been examined in response to your upcoming comments on the discussion section. Please refer to that specific section later in this document.

8. Lines 172-175: Authors mention seasonal variability and regional phenomena (e.g., ENSO events) but does not describe specific patterns or results. It is not reasonable to ask readers to infer patterns from figures or videos. Providing a detailed explanation would greatly improve the study's readability.

→ Regarding your comment #6, which we agree with, we believe it's important to avoid misleading the reader about the study's focus which is on decadal to climatic trends rather than seasonality. Mentioning seasonal variability could create confusion. To maintain clarity and stay focused on the paper's central message, we have removed the sentence.

9. Lines 214-218 and Figure 3: Why do neighboring provinces (e.g., 405 and 203) exhibit opposite patterns in average latitude evolution? This requires further explanation in the Discussion section.

→ Please refer to the responses to your comments on the Discussion section.

10. Line 217: Authors consider a ±0.5° poleward shift over the study period as "stable." Please provide evidence or references to support this criterion.

➔ This criterion was chosen based on the limited magnitude of change in mean latitude observed for provinces within this range, as well as the inherent variability of the dataset over the 26-year period. While no explicit reference establishes this exact threshold, our definition aligns with our goal of identifying provinces that exhibit the least latitudinal drift over time. Actually, provinces in that range represent the 20% of provinces undergoing the least latitudinal drift. We acknowledge that this definition is subjective and may vary depending on the context or study. However, in this analysis, it serves as a practical boundary for distinguishing stable provinces from those undergoing more pronounced shifts.

➔ Section 3.3, last sentence, l281: *"This range encompasses the 20% of provinces, exhibiting the least latitudinal drift over time, distinguished from the ones undergoing more pronounced and meaningful drifts."*

Discussion section: The results are interesting, but the discussion lacks depth. I suggest the following improvements:

Lines 270-271: Further discussion is needed to explore why the area has shrunk. Based on the results of this study, what are the main environmental variables driving the reduction? How do changes in those three environmental variables lead to the shrinking of provinces? Similarly, why have some areas expanded? This is unlikely to be a simple consequence of spatial redistribution. A deeper analysis could provide valuable insights for climate change policy-making.

➔ To better characterize the provinces in terms of environmental variables, we conducted a new diagnostic. For each province, we analyze the data distribution for the three variables used in the clustering: averaged epipelagic layer temperature, stratification, and NPP. The box-plot figure (Figure 2) illustrates the data distribution of these three variables from 1998 to 2023, spatially averaged for each biome. The data considered are monthly values (to be faithful with the input data of the clustering, and the monthly provinces derived) of T, Str and NPP spatially averaged over each biome (i.e. one value of each environmental variable per month per biome). The boxplots show the median of data distribution in the rectangles' centers, top and bottom of the rectangles represent first and third quartiles, segments' ends represent percentiles 5 and 95, and the orange dots represent the outliers. Boxplots with distinguishable median values (either the greatest or the weakest for instance) and narrower ranges (meaning weak data variance) highlight the environmental variable most likely to characterize the province specificity.

➔ In the results section (3.1), we included the boxplot figure along with a detailed characterization of provinces and biomes based on environmental variables. In the discussion section, we analyze the primary environmental drivers for each biome and explore how changes in these variables contribute to variations in the surface area and latitudinal shifts of the provinces.

➔ Results, Section 3.1:

*"The different biomes, and associated provinces, are characterized by specific environmental regimes (Figure 2). Focusing on the biophysical conditions for each province, we consider the data distribution for averaged epipelagic layer temperature, stratification, and NPP. Figure 2 shows monthly values of these three variables from 1998 to 2023, spatially averaged for each biome.*

[Figure]

*Figure 2: Characterization of biophysical biomes using monthly environmental forcings: temperature of the epipelagic layer (T, °C), stratification (Str, °C), and NPP (mgC/m²/day) from 1998 to 2023. The analysis uses monthly values of T1, Str, and NPP, spatially averaged across each biome (i.e., one value per month for each environmental variable per biome). The boxplots depict the data distribution, with the median shown at the center of each rectangle, the first and third quartiles represented by the top and bottom edges of the rectangles, the whiskers extending to the 5th and 95th percentiles, and orange dots indicating outliers.*

Biome 1 (the tropical biome) is characterized by the warmest and most stratified waters, associated with relatively low biological production. A similar but less pronounced pattern is observed for Biome 2 (the subtropical biome). Biome 3 (the eastern boundary coastal upwelling systems) is by far the most

productive biome. Biome 4 (the oceanic mesotrophic systems) also exhibits high NPP values, though weaker than Biome 3. Biome 5 (the sub-polar biome) is weakly stratified, characterized by cold waters, and shares a similar NPP range with Biomes 1 and 2. Biome 6 (the polar biome) features the weakest stratification and the lowest epipelagic layer temperatures among all biomes.

➔ The relative roles of environmental variables in surface and latitude variations have been examined in response to your upcoming comment.

➔ Discussion: The whole discussion section has been revised and improved. The full paragraph is quoted at the very end of the document. *

Lines 273-275: I suggest the focus of the discussion should be shifted to the characteristics of productive provinces and subpolar provinces, providing a detailed explanation of the reasons behind their expansion. This section could be combined with the discussion in the previous paragraph. Additionally, sufficient literature references are needed to support the arguments with robust evidence.

➔ First, the characteristics of every province, including productive provinces and subpolar provinces, have been documented with the new boxplots characterization diagnostic (Figure 2). Please refer to the above response regarding this aspect.

➔ Second, from this biophysical characterization, the main environmental variables driving the provinces' spatial variations have been identified. Discussion section, paragraph 2, l346: *"Despite the complexity of the multifactorial causes behind the spatial variation of the provinces, we can attempt to infer the main environmental variables driving the provinces' spatial evolution and how changes in these variables contribute to their size variation and latitude drifting. Based on Figure 2, boxplots with median values particularly different from the others (e.g., either the highest or lowest) and narrower ranges (indicating low data variance) highlight the environmental variable most likely to characterize a province's specificity. Equatorial provinces (Biome 1) are characterized by very high temperature and stratification values, with narrow boxplots indicating weak data variance. Thus, temperature and stratification seem to be the most explanatory variables for these provinces. A similar but less pronounced pattern is observed in subtropical provinces (Biome 2). Productive provinces (Biomes 3 and 4) are highly distinguishable by their significantly elevated NPP values compared to others, suggesting that NPP is the most explanatory variable for these regions. Subpolar and polar provinces (Biomes 5 and 6, respectively) are marked by low stratification and cold waters, with polar provinces showing the weakest values among all biomes. Therefore, temperature and stratification appear to be the key explanatory variables for subpolar and polar provinces. "*

➔ Third, we included in the Supplementary Material (Figure S1) annual time series of the three variables (i.e., epipelagic layer temperature, stratification, and NPP), spatially averaged at the global scale. This displays how the global mean values of these environmental drivers vary over time, providing additional context for the discussion on the factors influencing surface and latitude changes.

Section 2.1., l116: *"Annual time series of these three variables (i.e., epipelagic layer temperature, stratification and NPP), spatially averaged on a global scale, are provided in the Supplementary Material (Figure S1). This illustrates how the global mean values of temperature, stratification, and NPP fluctuate over time, reflecting interannual variability and decadal trends at the global scale."*

Supplementary material: *"Figure S1: Annual time series of global mean epipelagic layer temperature T (°C, top panel), stratification of the mesopelagic ocean Str (°C, middle panel) and NPP (mgC/m2/day, bottom panel) from 1998 to 2023 (spatially averaged over the whole domain). "*

[Figure]

Section 4 Discussion: We reference Figure S1 and discuss its insights in the following paragraph. Please refer to the next point for the citation.

[revised manuscript text omitted]

Lines 280-282: The results are highly promising; however, the discussion is overly simplistic and superficial. A more thorough comparison with other studies is needed. Based on the findings of this study, the reasons behind poleward migration should be clarified. While the discussion briefly mentions temperature changes, this study also considers NPP. Are there any interactive effects between these two factors? Furthermore, the potential connections between changes in external environmental variables and micronekton should be explored. For instance, how does micronekton respond to temperature changes, what are the underlying mechanisms, and how does it influence the variation of ocean province definition?

➔ Please refer to "Fourth" in the above response (i.e., Discussion section, paragraphs 4 and 5) for the clarification of the reasons behind provinces' poleward migration.

➔ A bibliographic effort has been conducted to document the evolution of temperature and NPP changes in time, and the interactive effects between these variables. Please refer to "Third" in the above response (i.e., Discussion section, paragraph 3).

➔ A comparison with Ariza et al. 2022 is conducted for the evolution on provinces' surface in time, according to the biophysical characteristics of the provinces: Discussion section, paragraph 7, l410: *"In Ariza et al. (2022), the authors derive acoustic provinces from a*

*clustering using acoustic data as a proxy of micronektonic biomasses, which they reconstructed from biophysical data (satellite-derived chlorophyll concentration, sea surface temperature and subsurface dissolved oxygen). In Albernhe et al. (2024), our biophysical provinces were compared to these acoustic provinces, revealing a strong overall agreement, particularly in terms of latitudinal patterns, dynamic regions, and upwelling areas. However, our clustering method did not capture oxygen-driven patterns. Ariza et al. (2022) also explored the spatio-temporal variability of provinces in future projections extending to 2100. They predict a contraction of upwelling and subpolar provinces alongside an expansion of subtropical and temperate provinces, leading to a global decline of pelagic fauna. Despite the differing timeframes of the two studies (1998–2023 in our case vs. projections for 2000–2020 and 2080–2100 in Ariza et al.), their conclusions align with ours regarding the trends observed in these biogeographical provinces, and the consequences on mid trophic biomass."*

➔ A comparison with other bibliographic references is conducted for the general trend in micronekton biomass evolution. Discussion section, paragraph 7: *"This potential trend for micronekton biomass evolution on the historical period would be in range with studies on micronekton biomass climate projections (Bryndum-Buchholz et al., 2018; Kwiatkowski et al, 2019; Lotze et al, 2019; Tittensor et al., 2021; Ariza et al., 2022)."*

➔ A comparison with other bibliographic references is conducted for the general trend in species latitudinal migrations.

Discussion section, paragraph 7, l420: *"To end up regarding the poleward drifting of provinces, this valuable observation in range with the literature (Hastings et al., 2020; Pinsky et al., 2020), suggesting a potential poleward migration of micronektonic populations induced by temperature changes."*

➔ The potential links between the potential connections between changes in external environmental variables and micronekton are briefly mentioned, but the authors deliberately chose not to explore this in detail, as they believe it is beyond the scope of the study without a proper analysis of the temporal consistency of micronekton characteristics within the provinces.

Discussion section 4, l399 : *"If provinces characteristics are preserved over time, the shrinking of productive and polar provinces together with the expansion of the equatorial and tropical provinces would imply a global decline of micronekton biomass. This reasoning is based on a basic deduction from the consequences of ecological niche surface variation. However, the underlying mechanisms that could explain a global decline in micronekton biomass may be partly attributed to the previously mentioned decreasing trend in NPP at the global scale. Since NPP is at the base of the trophic chain, this decline has cascading effects up to micronekton, limiting their energy sources and thus reducing population development. Additionally, the potential global decline in micronekton biomass may also be partly induced by the increasing trend in global ocean temperature, which affects micronekton development times (Gillooly et al., 2002), including growth and mortality."*

A thorough analysis of the temporal consistency of micronektonic characteristics within the provinces, such as biomass and vertical structure, would first need to be conducted. This would help confirm that these features remain stable over time within the provinces, thereby validating the relationship between changes in province surface areas and global micronekton biomass. Such an analysis would require a separate, dedicated study. Without evidence of the temporal stability of micronektonic features, any exploration of underlying mechanisms would lack a solid foundation and could lead to unreliable conclusions. We draw the attention to this point in the last paragraph of the discussion :

*"Finally, a detailed analysis of the consistency of each province's micronektonic characteristics over time (regarding biomass and the vertical structure of micronekton), should be conducted. This would provide a more solid basis for confirming the link between changes in province surface areas and the evolution of total micronekton biomass."*

➔ Reference added to the reference section : Gillooly, J. F., Charnov, E. L., West, G. B., Savage, V. M., & Brown, J. H. (2002). Effects of size and temperature on developmental time. Nature, 417(6884), 70-73.

Technical corrections:

Line 18: Avoid including references in the Abstract.

➔ Sentence rephrased to remove the reference : *"A new method has been proposed in the literature to define provinces that identify micronekton functioning patterns based on environmental variable."*

Line 111: Specify "stratification" of what?

➔ Specified l78: *"stratification of the mesopelagic ocean temperature"*

➔ Vocabulary clarification L106: "the temperature gradient between the epi and the meso-pelagic layers, as an index of the stratification (hereafter referred to as 'stratification')"

Figure 1: The color area for "602" is unclear. Only the number is visible in the top right.

➔ The polar province 602 is a seasonal province, expanding only during winter and disappearing entirely in summer. In the average ocean state represented by the reference biophysical biomes (Fig. 1), this province is also not visible. Therefore, we chose to retain its numbering to indicate the location of its seasonal occurrence.

Line 222: Correct "VGMP" into "VGPM"

➔ Done.

Line 231: Please ensure consistent section numbering. Change "II.1" to "2.1."

➔ Done.

Please provide the citation for the Albernhe et al., (2024) if it has been published.

➔ Added to the reference section:

[revised manuscript text omitted]

---

## Author Comment (AC2)

**Response to Referee #2 comments**

We sincerely appreciate the reviewer's constructive comments and valuable suggestions, which have contributed to enhancing the clarity and robustness of our manuscript. In this document, the reviewer's comments are presented in black, while our associated responses are presented in blue. Additionally, we provide the revised version of the manuscript.

The paper titled "Micronekton Indicators Evolution Based on Biophysically Defined Provinces" by Albernhe et al. examines the evolution of micronekton indicators across biophysically defined provinces over a 25-year period, from 1998 to 2023.

This study offers a dynamic, long-term analysis of micronekton provinces, providing quantitative insights into surface area and latitude trends while presenting direct evidence of climate-driven shifts in marine ecosystems. It builds upon previous research by offering a more precise, time-resolved, and geographically validated understanding of micronekton variability.

By focusing on two key indicators—surface area trends and average latitude shifts—this study highlights metrics that could serve as future Ocean Monitoring Indicators. Its straightforward and impactful contribution makes it highly relevant for the OSR.

My primary concern is that this paper heavily relies on the work by Albernhe et al. (2024), which is still under review and not yet publicly available. Additionally, no reference to this paper is provided, leaving significant gaps in the current contribution. Without the publication of Albernhe et al. (2024), too many critical questions remain unanswered.

Therefore, unless Albernhe et al. (2024, under review) is officially published, I must recommend rejecting this submission. However, given the relatively long publication process for OSR9, this contribution could be accepted on the condition that the referenced paper is published before the final release of OSR9.

Furthermore, I suggest revisions, and the authors should provide a copy of Albernhe et al. (2024, under review) alongside their revisions to allow for a thorough evaluation of the current submission.

➔ Thank you for your thoughtful feedback. We appreciate your concern regarding the reference to the unpublished paper and the importance of providing sufficient context for our study. We are pleased to inform you that the referenced study, "Albernhe et al., 2024," has since been accepted and published. The full reference is as follows:

Albernhe, S., Gorgues, T., Lehodey, P., Menkes, C., Titaud, O., De La Giclais, S. M., & Conchon, A. (2024). Global characterization of modelled micronekton in biophysically defined provinces. Progress in Oceanography, 229, 103370.

We have updated the manuscript to reflect the status of this publication (i.e., Introduction l73, l93, Material and methods l100, l103, l116, etc.), and added the reference in the appropriate section.

However, we want this paper to be comprehensive on its own. The different steps of the methodology are detailed in Section 2.1., l102 : "While the overall methodology is detailed in Albernhe et al. (2024), we outline the different steps of the method below to ensure this study is comprehensive and self-contained. [...]"

**Major comments**

1) It remains completely unclear how the 27 provinces are obtained. I suggest that authors will provide more information about the determination of 27 provinces in Materials and Methods section. Otherwise, the paper cannot be read as stand alone paper.

➔ We appreciate the reviewer's concern regarding the determination of the 27 provinces. We have ensured that this process is clearly explained in the Materials and Methods section. While we followed the methodology of Albernhe et al. (2024), we dedicated a specific section to detailing our approach (Section 2.1).

➔ First, we derived six biophysical biomes through the clustering of environmental variables (see Section 2.1 paragraphs 4 and 5, lines 121–133). Then, each biome was subdivided into provinces based on hemisphere and ocean basin, resulting in 27 provinces as subdivisions of biomes (see Section 2.1, lines 143): *"The six biophysical biomes obtained from the clustering of environmental data characterize homogeneous environmental regimes on a global scale. Since similar oceanographic regimes occur in multiple locations, biophysical biomes extend across various ocean basins. In this study, we also delineate "provinces" as subdivisions of biomes at the scale of ocean basins and hemispheres that have been shown to be characterized by stable biophysical drivers and potential taxonomic identity (Spalding et al., 2012; Sutton et al., 2017; Albernhe et al., 2024). This subdivision of each of the six biophysical biomes results in the definition of 27 provinces, establishing regional frameworks for studying micronekton."*

➔ To improve clarity, we have revised these paragraphs to better explain our methodology. We hope this addresses the reviewer's concern and enhances the manuscript's readability as a stand-alone paper.

2) The manuscript will benefit if more in-depth discussion in comparison to the works by (Reygondeau et al., 2012), (Sutton et al., 2017), (Costello et al., 2017), (Elizondo et al., 2021), (Proud et al., 2017), (Ariza et al., 2022) is provided.

➔ The Discussion section has been significantly improved, notably by deepening the comparison with several studies. Additionally, substantial work has been done to better contextualize the findings, expand the bibliography, and refine the interpretation of the results since the initial version.

Section 4, Discussion, paragraph 1, l329: *"In this study, we defined 27 biophysical provinces linked to micronekton, based on a methodology introduced in Albernhe et al. (2024). Our definition of the reference biophysical provinces (Figure 1) has been compared to other studies (e.g. Proud et al., 2017; Sutton et al., 2017, Ariza et al. 2022) employing comparable methodologies using environmental variables to derive biogeographic regions. Sutton et al. (2017) classified regions based on environmental drivers and expert knowledge, Proud et al. (2017) used clustering on environmental variables to model deep scattering layers characteristics, and Ariza et al. (2022) derived provinces by clustering acoustic data reconstructed from biophysical variables. Despite methodological differences among these three studies, the resulting biogeographical regions closely align with ours. Notably, beyond the evident latitudinal banding (in the austral Ocean for instance), more complex regional structures emerge in the North Atlantic, midlatitude frontal zones (except in the South Pacific), and upwelling regions (see Figure 4 in Sutton et al., 2017; Figure 3A in Proud et al., 2017; Figure 2a in Ariza et al., 2022; and our Figure 1). This similarity likely arises because all approaches rely on*

*biophysical variables that capture key information on temperature, biological productivity, and water column mixing."*

➔ Section 4, Discussion, line 410: *"In Ariza et al. (2022), the authors derive acoustic provinces from a clustering using acoustic data as a proxy of micronektonic biomasses, which they reconstructed from biophysical data (satellite-derived chlorophyll concentration, sea surface temperature and subsurface dissolved oxygen). In Albernhe et al. (2024), our biophysical provinces were compared to these acoustic provinces, revealing a strong overall agreement, particularly in terms of latitudinal patterns, dynamic regions, and upwelling areas. However, our clustering method did not capture oxygen-driven patterns. Ariza et al. (2022) also explored the spatio-temporal variability of provinces in future projections extending to 2100."*

**Detailed comments**

L172-174 The following statement is too general "Seasonal variability can be observed with the latitudinal shifts of the horizontal boundaries, as well as regional seasonal phenomena or isolated phenomena like ENSO events." Without any hints on what to look from the video, it requires detailed analysis of the video itself by readers.

➔ We agree with your comment, and we believe it's important to avoid misleading the reader about the study's focus which is on decadal to climatic trends rather than seasonality. Mentioning seasonal variability could create confusion. To maintain clarity and stay focused on the paper's central message, we have removed the sentence.

Section 3.4 More appropriate title is Sensitivity analysis instead of Uncertainty. Text of the section should be adjusted accordingly

➔ Section 3.4 has been titled accordingly.